# Effect of Tool Offset on the Microstructure and Properties of AA6061/AZ31B Friction Stir Welding Joints

**Huachen Liu [1], Yikun Chen [1],\*, Zhenhua Yao [2],\* and Feixiang Luo [2]**

[1]  New tobacco products Engineering Center, China Tobacco Hubei Industrial Co., Ltd., Wuhan 430040, China; Wangcf94@163.com

[2]  School of Material Science and Engineering, Wuhan University of Technology, Wuhan 430070, China; m13628654382_1@163.com

\*  Correspondence: g123-1@163.com (Y.C.); zhyao@whut.edu.cn (Z.Y.); Tel.: +86-13720316080 (Y.C.); +86-13871171994 (Z.Y.)

**Abstract:** Friction stir welding was carried out on AA6061/AZ31B alloy, and the influence of tool offset on microstructures and mechanical properties of the joints was studied. The results showed that preferred properties were obtained when Mg was placed in the Advancing Side (AS) and offset was positioned into Mg. The distribution of $Al_3Mg_2$ and $Al_{12}Mg_{17}$ intermetallic compounds (IMCs) could be improved with offset in a certain range. The tensile strength of the joints was elevated with the increase of the offset due to the superior distribution and diminished size of IMCs, and when the offset was 1.5 mm, the joint strength reached the maximum value of 107 MPa. The microhardness of the stirring zone decreased with the increased offset.

**Keywords:** aluminum alloy; magnesium alloy; friction stir welding; microstructure; tensile strength

## 1. Introduction

With the increasing demand for lightweight material structures, new technologies are constantly being developed and applied in manufacturing industry in order to save weight and costs. Dissimilar materials welding can effectively combine the advantages of two different metals [1,2] and obtain composite structure with light weight. Recently, magnesium and aluminum alloys are used extensively in the aerospace and electronics industries due to their excellent properties such as high specific strength, good corrosion resistance [3,4], and electromagnetic shielding properties [5]. However, there are several problems in welding of Al/Mg alloy such as: (i) large thermal stress in welding which would lead to hot cracks, (ii) unstable metallurgical bond results from difference of lattice types, and (iii) intermetallic compounds (IMCs) formed at high temperatures which would seriously deteriorate the properties of the joints [6,7].

Friction stir welding (FSW) is a new solid-state joining technique, which could produce reliable joints of dissimilar alloys owning to its relatively low and controllable heat input and enough interface contact between the two materials [8,9]. In this process for dissimilar alloys, effective and comprehensive mutual material flow is more indispensable for joint with ideal quality compared to the combination of similar alloys. Therefore, besides the rotational speed and welding speed, research focused on the position of base materials, tool offset and transferred behavior of materials have been performed and reported continually recently [10,11]. Sato [12] found that when AZ31 magnesium alloy was placed on the advanced side (AS) and 1050 aluminum alloy was placed on the retreating side (RS), the joints with better forming quality could be obtained, while Mclean [13] reported that when 5083 aluminum alloy was placed on the AS, the joints appeared high quality. Buffa [14] found that FSW of

different materials would be more effective when the softer material is placed in the RS based on the experiments and simulation of the process of aluminum and magnesium alloy. Yamamoto [15] found that the obtained welds had cavity defects or surface flash when the pin was offset to either AS or RS. However, Firouzdor [16] found that high strength joints were obtained only when the magnesium alloy was chosen as the AS and the pin was placed in this side. Kailas [17] reported that there was a window of interface position in relation to the tool in the AS which could give the optimal strength and ductility.

In another side, in dissimilar welding controlling the temperature is also extremely important as material flow in order to avoid the formation of intermetallic compounds which could adversely influence the weld metal properties. However, the effect of tool offset on the characteristics of the IMCs have less been reported.

Thus, in this paper, FSW was performed to join AA6061/AZ31B alloy, and the effects of tool offset on microstructures, especially in characteristics of IMCs and the mechanical properties of the joints were systematically studied.

## 2. Experimental Procedures

The chemical compositions of AA6061-T6 Al and AZ31B-O Mg alloy are listed in Table 1, and they were both cut into the size of 250 mm × 100 mm × 6 mm. Before FSW process was conducted, the oxide film of the alloy was removed and the surface was cleared by acetone.

**Table 1.** Compositions of Al and Mg alloys (wt%).

| Element | Si | Fe | Cu | Mn | Zn | Cr | Ni | Mg | Al |
|---------|------|--------|--------|------|------|------|---------|-----|-----|
| AA6061 | 0.537 | 0.28 | 0.12 | 0.16 | 0.11 | 0.17 | - | 0.56 | bal |
| AZ31B | 0.05 | 0.0022 | 0.0018 | 0.35 | 0.82 | - | 0.00016 | bal | 3.2 |

The welding was performed on FSW-TS-S08 friction stir welding machine. The tool was made of H13 tool steel, and the pin was tapered with thread. The shoulder was concave-shaped with the diameter of 15 mm, root diameter 7.6 mm, tip diameter 5 mm, and pin length 5.6 mm. The angle of tilting between the tool and workpiece was 3° and the plunge depth was 0.3 mm. Room temperature is ranged between 17 °C and 25 °C when the process was carried out.

To choose the proper one placed in Advancing Side (AS), the surfaces of the joint obtained with Al and Mg as AS material respectively were compared, as shown in Figure 1. There was an obvious zigzag line along the direction of the welding and the surface was rough when Al alloy was placed in AS, so in all following experiments in this study, Mg alloy was placed in AS.

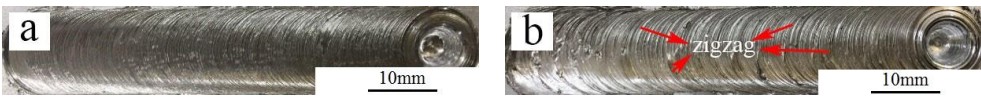

**Figure 1.** Surface morphologies of weld with AS: (**a**) Mg, (**b**) Al.

FSW was carried out on the equipment (FSW-TS-S08, FSW Co., Wuhan, China), and other parameters of FSW are shown in Table 2. The rotation speed was 550 rpm, and welding speed was 15 mm/min. 7 different tool offsets were selected: 0.5 mm to Al, no offset, 0.5 mm, 1 mm, 1.5 mm, 2 mm, and 2.5 mm to Mg, respectively. To distinguish between the different specimens, they were labeled from T1 to T7.

Samples with size of 10 mm × 10 mm × 6 mm were taken from the center of the joints for X-ray diffraction (XRD, D8 Advance, Bruker, German) analysis with Cu Kα and scanning angle between 10° and 90° and for Electron Probe Micro Analysis (EPMA, JXA-8230, Tokyo, Japan). The tensile specimens were cut from the direction perpendicular to the welding direction. Five standard ASTM E8-M specimens from T1 to T7 were taken from the welding part. A tensile test was conducted

on an electronic universal material testing machine (Instron 1341, Instron, Boston, MA, USA) with 1 mm/min crosshead speed. Tensile fracture was observed by SEM (JSM-IT300, JEOL, Tokyo, Japan). The microhardness test was performed on a microhardness tester (HV1000, LunJie, Shanghai, China) with 200 g load and 10 s dwell time. Test points were located in the middle of the plate thickness and distributed symmetrically on the surface of each specimen with an interval of 1 mm.

**Table 2.** Numbered samples with different tool offset.

| Specimens | Rotation Speed (rpm) | Welding Speed (mm/min) | Material of AS | Tool Offset (mm) |
|---|---|---|---|---|
| T1 | | | | 0.5 to Al |
| T2 | | | | 0 |
| T3 | | | | 0.5 to Mg |
| T4 | 550 | 15 | Mg | 1.0 to Mg |
| T5 | | | | 1.5 to Mg |
| T6 | | | | 2.0 to Mg |
| T7 | | | | 2.5 to Mg |

## 3. Results and Discussion

### 3.1. Macromorphologies of the Joints

The surface and cross-section morphologies of the welds with different tool offset were exhibited in Figures 2 and 3. AS and RS were lablled in Figures 2T1 and 3T2, other graphs were same with them respectively. There were defects both on the surface and cross-section when the offset was zero or in aluminum, as seen in Figure 2T1,T2 and Figure 3T2. However, when the tool was offset to Mg alloy, preferred weld quality was obtained There were two aspects related to the production of the defects: materials flow and IMCs. Because the friction between Al alloy and tool was greater than that of Mg alloy [18], when the tool is located in the side of aluminum, heat generated was larger. This would lead to more formation of IMCs, which could lead to hot crack in the following cooling [19]. In another side, the plastically deformed material at the AS flowed in counter-clockwise direction and passed the tool at the RS under the shear of the tool, which would push the materials at the RS to move behind to fill the empty position left by the tool moving ahead. This materials transfer was important for desired joint. Mg alloy has relative higher shear strength and lower ductility compared with Al alloy due to its HCP structure [18], so when no tool offset was placed in Mg, the transferred materials motivated by the tool were probably not enough, so cavities and tunnel defects appeared due to the insufficient material flow. However, when the tool offset reached 2.5 mm for T7, the center of the pin would deviate far from the butt surface, which could result in release of stirring effect and decrease the flowing extent of the welding materials, thus microvoid would appear. Moreover, the shape of the pin used was conical, so under the condition of large tool offset, the material at the bottom of the butt surface was not stirred, which would lead to the formation of gap at the bottom of the weld.

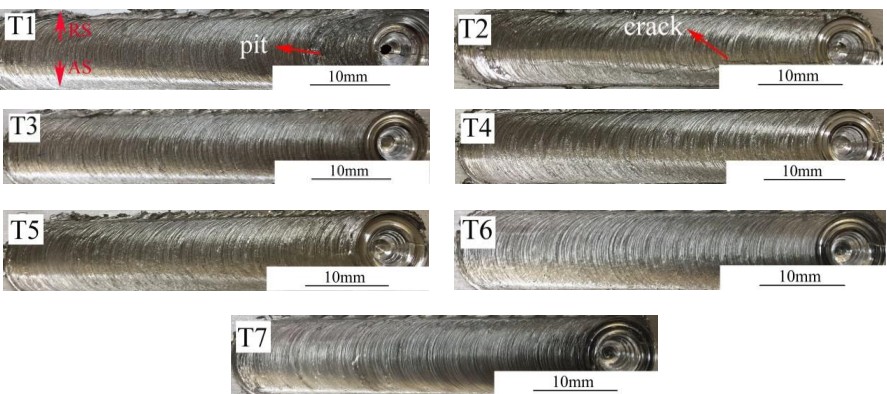

**Figure 2.** Surface morphology of the welds at different tool offset.

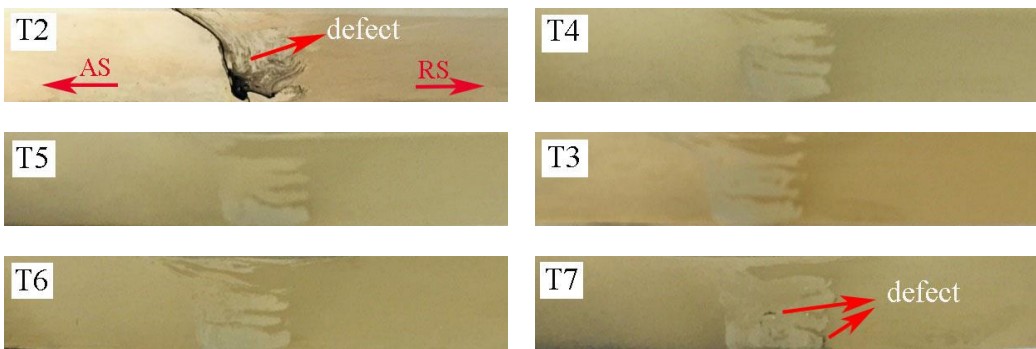

**Figure 3.** Morphologies of cross-section with different tool offset.

### 3.2. Microstructure of the Welds

XRD patterns of the weld zone of T2–T6 are shown in Figure 4, $Al_3Mg_2$ and $Al_{12}Mg_{17}$ IMCs were both observed. However, with the increase of offset, the diffraction peak of $Al_3Mg_2$ disappeared gradually, while that of $Al_{12}Mg_{17}$ changed little. The formation of IMCs indicated that the temperature had exceeded the melting point of the metals in the stirring zone [18], even the offset to Mg have elevated the translating temperature because of the increased content of Mg in that zone. However, Stirring zone (SZ) would move toward Mg matrix side when the offset was present, so the relative content of Al decreased in this area, which would be in favor of formation of $Al_{12}Mg_{17}$.

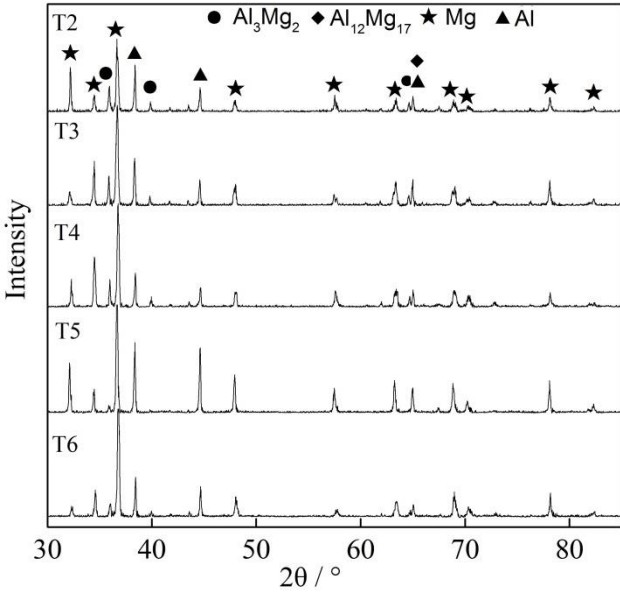

**Figure 4.** XRD patterns of the weld zone with different tool offset.

Morphologies of interface between the Mg alloy base material and the SZ of T2, T5, and T7 were exhibited in Figure 5, and EDAX analysis of the corresponding phases are shown in Table 3. Cracks were observed at the interface and a large number of Al band appeared on the right side of the interface when the offset was absent, as seen in Figure 5a. When the tool was offset to Mg alloy with 0.5 mm, cracks disappeared and the size of band structures was reduced with more homogeneous distribution. Al alloy at the RS was deformed by the periodic stirring and appeared layer by layer distribution, this was consistent with the adhesion transfer mechanism [20]. There was an obvious gap between the interface line and the region where Al band was observed when offset was present, as seen in Figure 5b. Interface line between Mg and SZ indicated the trajectory of outer edge of stirring tool. When the tool moving forward, empty position was produced behind and would be filled instantaneously

by surrounding materials under compression or shearing, as described in Figure 6. Meanwhile, the amount of Al transferred into the Mg matrix decreased because of the offset, and the moving of Al was mainly driven by the shearing of stirring, which would limit the amount and displacement of Al transferred to interface region compared to compress force. When the offset reached 2.5 mm, Al band was further away from the interface, so the region for EPMA analysis was chosen toward the middle line, as seen in Figure 5c. EPMA indicated that IMC was formed in the stirring process, and $Al_3Mg_2$ was mainly distributed along Al band, while $Al_{12}Mg_{17}$ was granular and observed in the Mg matrix. The kinds of precipitated IMC was mainly depended on the local composition of the elements, $Al_3Mg_2$ would tend to be formed in the region rich in Al, so it was mainly close to Al band. $Al_{12}Mg_{17}$ was formed with the combination of Mg matrix and small Al particle or band, which was also formed in the stirring process, so it was granular and distributed relative homogeneously in the matrix.

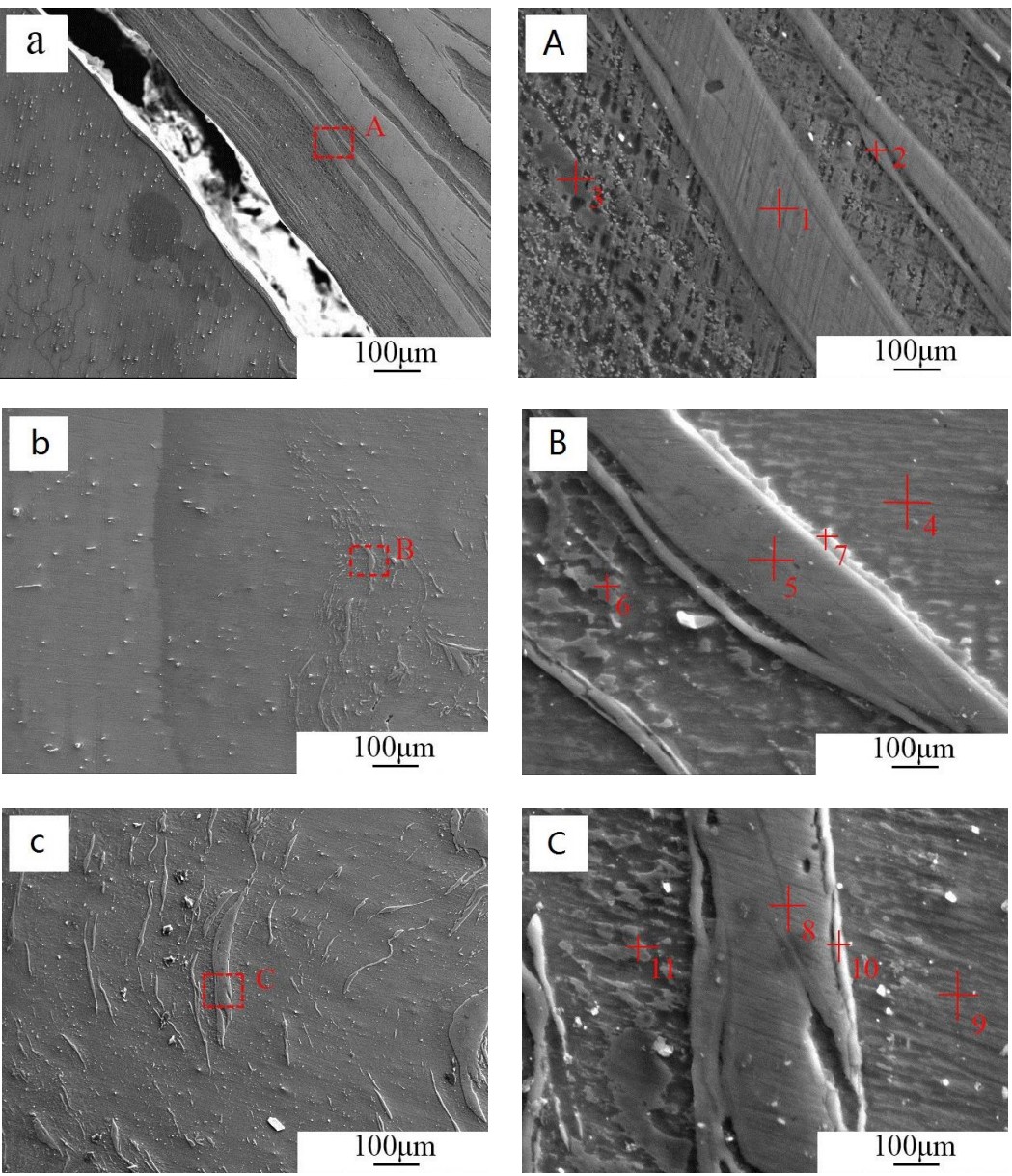

**Figure 5.** Morphologies near the interface of the sample: (**a**) T2, (**b**) T5, (**c**) T7,(**A**) magnification of T2, (**B**) magnification of T5, (**C**) magnification of T7.

**Table 3.** EPMA analysis of T2, T5 and T7 (wt%).

| Points | Mg | Al | O | Si | Possible Phase |
|---|---|---|---|---|---|
| 1 | 2.53 | 96.08 | 0.86 | 0.52 | Al |
| 2 | 30.91 | 68.09 | 1.00 | 0 | Al + Al$_3$Mg$_2$ |
| 3 | 94.68 | 3.43 | 1.36 | 0.53 | Mg |
| 4 | 95.34 | 3.61 | 1.05 | 0 | Mg |
| 5 | 2.74 | 95.47 | 1.12 | 0.67 | Al |
| 6 | 59.79 | 39.46 | 0.74 | 0 | Mg + Al$_{12}$Mg$_{17}$ |
| 7 | 37.08 | 62.92 | 0 | 0 | Al + Al$_3$Mg$_2$ |
| 8 | 3.13 | 95.36 | 0.93 | 0.58 | Al |
| 9 | 95.34 | 3.61 | 1.05 | 0 | Mg |
| 10 | 28.33 | 70.74 | 0.93 | 0 | Al + Al$_3$Mg$_2$ |
| 11 | 61.67 | 37.52 | 0.81 | 0 | Mg + Al$_{12}$Mg$_{17}$ |

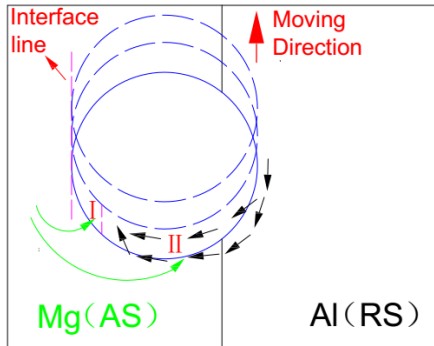

**Figure 6.** Materials flow sketching in stirring process with offset.

### 3.3. Microhardness of the Joints

The microhardness distribution along the line perpendicular with the butt surface of T2–T7 are shown in Figure 7. The stirring zone were in the middle, and Al matrix and Mg matrix were on left and right sides respectively. The hardness of the stirring zone decreased gradually with increasing offset toward Mg side. When the amount of tool offset was 0 mm, 0.5 mm and 1.0 mm, the microhardness of the stirring zone was much higher than that of both Al and Mg alloy. However, when the amount of tool offset reached 1.5 mm, the hardness of the stirring zone were greater than that of Mg alloy, while lower than that of Al alloy. When the amount of tool offset was small, large number of continuous rough banded IMCs formed in the stirring zone which would increase the hardness. With the increasing offset toward Mg, the heat input was reduced, and this would reduce the amount of IMCs in the stirring zone. Meanwhile, the morphology of IMCs was changed to be short strip shape and was dispersed more homogeneous, these would both decrease the hardness of this area.

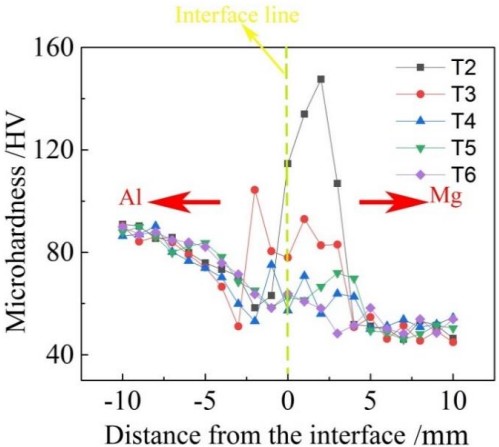

**Figure 7.** Microhardness distribution of the joints with different tool offset.

### 3.4. Tensile Properties and Fracture Analysis

The tensile strength of the joints with different offsets were exhibited in Figure 8. The strength increased first with increasing offset, and reached the maximum value of 107 MPa corresponding to T5. However, when the offset was elevated further, the strength decreased, and it was only approximatly 49 MPa for T7. When the offset was zero for T2, tunnel and cracks were produced in the welding, and the size, shape and distribution of IMC were all beneficial to the propagation of the cracks, then deteriorate the properties of the material under the tensile stress. With the increase of the offset, the defects disappeared and the IMCs exhibited superior distribution and size, which was in favor of increase of the strength of the joints. When the offset increased to 2.5 mm, the tensile strength of the joint decreased sharply, which was mainly caused by the alleviation of the stirring between the two metals, as seen in Figure 5c,C. The size of Al stirred into the Mg matrix become larger again due to the alleviated force such as friction, shearing and extrusion.

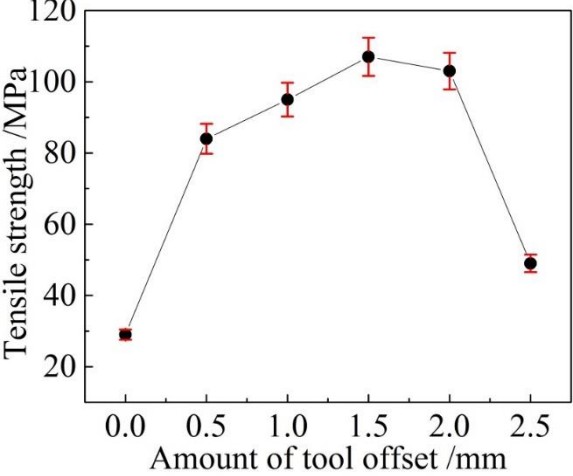

**Figure 8.** Tensile strength of the joints with different tool offset.

Morphologies of the tensile fracture of the joints with different tool offsets are exhibited in Figure 9. Crack defects could be observed in the fracture of T2, and river-like pattern and cleavage steps were appeared for all samples, so it was brittle fracture for FSW Al/Mg dissimilar alloy with or without offset, this was mainly resulted from the the existence of IMCs in the joints. It could be indicated by the elongation of the joint, which was less than 1%.

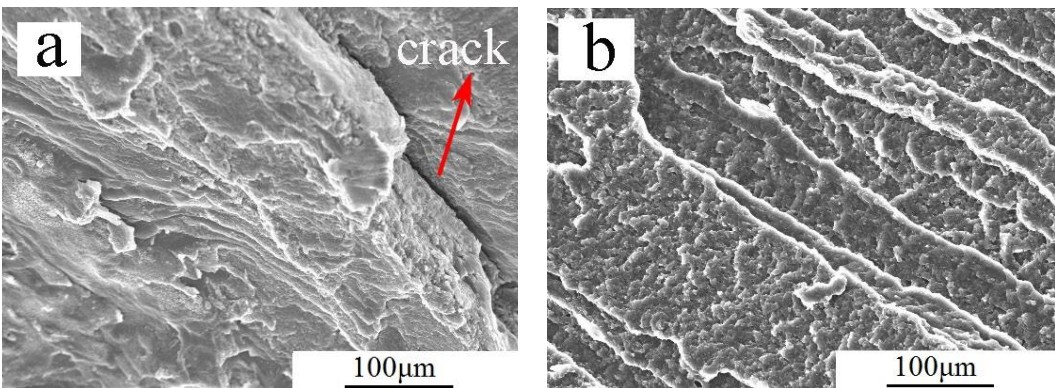

**Figure 9.** *Cont.*

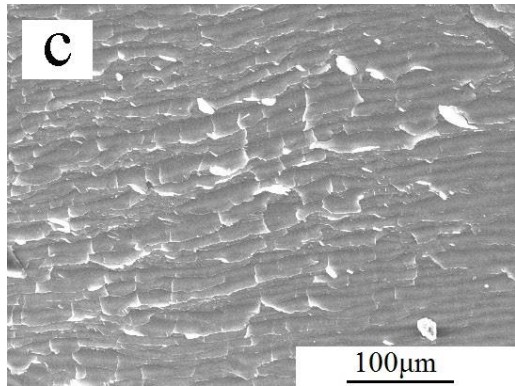

**Figure 9.** Morphologies of tensile fracture with different tool offset: (**a**) T2,(**b**) T5,(**c**) T7.

## 4. Conclusions

The effects of tool offset on microstructures and mechanical properties of Al/Mg dissimilar alloy FSW were investigated, and the conclusions could be listed as followed:

(1) Preferred properties were obtained when Mg was placed in the AS and offset was positioned into Mg. However, when offset reached 2.5 mm, defects were appeared due to the insufficient stirring process between Al and Mg. It meant that less plastic and ductile materials should be placed in AS and with a certain range of tool offset in FSW of dissimilar alloys.

(2) $Al_3Mg_2$ and $Al_{12}Mg_{17}$ IMCs could not be completely eliminated by offsetting the tool to Mg alloy. However, the distribution could be improved within a certain range.

(3) With the increase of the offset, the tensile strength of the joints increased first, and when the offset was 1.5 mm, the joint strength reached the maximum value of 107 MPa. The maximum microhardness of the stirring zone decreased with the increased offset.

**Author Contributions:** Conceptualization, Y.C. and Z.Y.; Methodology, H.L.; Experiments and Data, F.L.; Writing—original draft preparation, Z.Y. and F.L.; Writing—review and editing, Z.Y. and F.L.; All authors have read and agreed to the published version of the manuscript.

**Funding:** "This research was funded by Natural Science Foundation of China, grant number 51801140" and "The APC was funded by Natural Science Foundation of China".

**Acknowledgments:** This work was carried out with financial support from Natural Science Foundation of China (51801140), This work was also supported by Analytical and testing center in Wuhan University of Technology.

**Conflicts of Interest:** The authors declare no conflict of interest.

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
