# Peer review of "Effect of Tool Offset on the Microstructure and Properties of AA6061/AZ31B Friction Stir Welding Joints"

_metals, doi:10.3390/met10040546_

Round 1
Reviewer 1 Report
1.Introduction. This section has to be expanded to include discussing on the reasons behind the inconsistency of the already published results and how this paper might help to improve the situation. Otherwise, this research is cognized as another one to increase the inconsistency even more. What was the objective of this paper? Why such a study is necessary?
2. Experimental procedures. Information about manufacturers and their countries must be added to all equipment trademarks. Also information is lacking about XRD conditions such as radiation, exposure, wavelength , etc. How many samples were prepared for tensile testing?
3. Results and discussion. Section 3.1 must be totally rewritten for better clarity and consistency with the known FSW physics.
Page 3, line 84. ...greater than that of Mg alloy[16],... Ref.16 says nothing about friction coefficient difference between Mg and Al. It was just shown the temperature difference which may be provided by more intense exothemic reactions between Al and Mg.
Page 3, line 85. Successive heat input would lead to the increase of local liquid phase ... Liquid phase appears by exothermic reaction between Al and Mg and constitutional liquation.
https://doi.org/10.1080/13621718.2016.1248648
Page 3, lines 86-88. In another side, serious adhesion of the tool would occur at higher temperature, which resulted in serious metal loss in the weld zone, so pit defects appeared at the end of the weld, ... As far as I know there is not any loss of metal in the weld zone. What loss is meant by this?
Page 3, Figure 2. Please, identify both AS and RS as well as Al and Mg in these macrograph
Page 3, lines 98-101. Mg alloy has poor ductility due to its HCP structure, even 98 if the Mg alloy was placed in AS with strong stirring effect, the lack of deformability would still lead 99 to cavity defects....This statement is arguable since both metals are heated, deformed to nanograins and thus plasticized. Deformation mechanism in this state is quite different from translational dislocation glide.
Figure 3. Please, identify AS and RS as well as Al and Mg in the micrographs.
Page 4, lines 118-119.However, SZ would move toward Mg matrix side when the offset was present, so the relative content of Al was decreased in this area, which would be in favor of formation of Al12Mg17. Why then the Mg peaks have lower intensity in T7 as compared to those in T2?
Figure 4. Why all Mg peaks are lower in T7 as compared to T2? Using more offset to Mg it is anticipated that more Mg will be mixed with Al and therefore higher will be its XRD peaks like in case of T3 to T6. Please, identify all peaks using the Miller indexes. The X-axis is Theta or 2 Theta?
Page 4, lines 122-135. Such an explanation lacks the role of adhesion of metal to the FSW tool which plays an important role in the material transfer. Alternative mechanism of weld formation in FSW is known as layer-by-layer transfer by adhesion, see for example
S. Yu. Tarasov, A.V. Filippov, E.A. Kolubaev , T.A. Kalashnikova Adhesion transfer in sliding a steel ball against an aluminum alloy.Tribology International 115 (2017) 191–198
http://dx.doi.org/10.1016/j.triboint.2017.05.039
The rear zone behind the tool is a stagnant pocket so that no compression is there during constant moving of the FSW tool.
More detailed and physically based explanation is needed.
Page 5, line 136. EPMA or EDS cannot indicate the intermetallics directly since the electron probe spot may be too large and therefore give miixed chemical composition to include both intermetallics and base metals. Only XRD shows them directly.
Figure 6. This scheme explains nothing. How does Mg transfer from the AS to the rear zone in direction opposing th that of the FSW tool rotation?
Material transfer in FSW is not similar to that of viscous fluid because of its strong adhesion to the tool and interlayer cohesion.
S. Yu. Tarasov, A.V. Filippov, E.A. Kolubaev , T.A. Kalashnikova Adhesion transfer in sliding a steel ball against an aluminum alloy.Tribology International 115 (2017) 191–198
http://dx.doi.org/10.1016/j.triboint.2017.05.039
Fgure 7. The X-axis is denominated as "distance from the butt surface". It is confusing axis name which may be perceived as the "distance below the surface". It follows from the text that this is the microhardness number distribution in a weld cross section along the line parallel to the plate surfaces. Also a former centerline joint line must be shown in Fig.7 to evaluate the offset effect.
Figure 9. Sample must be identified in the Fig.7
A standalone "Discussion" section is lacking in this manuscript where all the results obtained should be generally discussed and compared with those from the literature sources. Therefore, this manuscript does not serve to improve the situation and conclusions look too partial.

Reviewer 2 Report
Grammar is understandable but could be improved, e.g. line 26: Recently, magnesium and aluminum alloy are used extensively
Some inconsistent formatting, e.g. around line 40 with names: A.A Mclean reported that when 5083 aluminum alloy was placed on the AS, the joints appeared high quality. N Yamamoto [14] found that the obtained welds had cavity defects or surface flash when the pin was offset to either AS or RS. However, Vahid[15]
Some questionable word choice, eg at line 43: when the magnesium alloy was placed on the AS and owned the pin’s offset. And line 85 Successive heat input and line 168 distribution of IMC were all benific to the propagation
No units for Table 1. Presumably weight %
Literature review could be expanded to expand the review of papers on this topic, for example
Positional dependence of material flow in
friction stir welding: analysis of joint line
remnant and its relevance to dissimilar metal
welding
K. Kumar and S. V. Kailas
Friction stir welding of dissimilar aluminium–
magnesium joints: sheet mutual position
effects
G. Buffa*, D. Baffari, A. Di Caro and L. Fratini
Should be a bit more general about conclusions, and attempt to draw at least one extrapolation from the results. For example the offset found to be optimum will be affected by the geometry of the tool used. The authors should make an effort to extrapolate this other tool geometries or alloy combinations.
Conclusion:
Nothing too surprising, but a good comprehensive examination of a topic of interest. Good reference material.
Reviewer 3 Report
Dear Authors,
the Research presented valuable results about the effect of tool Offset on the microstructural and mechanical behavior of hybrid friction stir welded Joints.
Unfortunately, the paper does not have novelty in the field of friction stir Welding. However, it can be a good conference paper.
Reviewer 4 Report
This work focuses on the impact of tool offset on the microstructure and mechanical properties of FSWed AA6061/AZ31B joints. The process parameters have been validated. The main novelty of this article have to be clarified in the introduction. The experimental procedure requires to be improved and some information should be added to the paper. The metallurgical features of the weld i.e. grain size measurement can be enhanced. The English language can be improved. To present a robust research article, the authors need to address the following comments.
Introduction
Line 26: Please use the plural form of alloys in “Recently, magnesium and aluminum alloy are…”
Line 29: The word “There” should be written in small letter in “However, There are several problems in welding…”
Line 31: The term “IMCs” should be introduced in the text.
Line 47&48: The authors mentioned that “In this paper, FSW was performed to join AA6061/AZ31B alloy, and the effect of tool offset on microstructures and mechanical properties of the joints was systematically studied.” From this sentence, the aim of this paper is not clear and the question is that what is the uniqness of this work compared to the previous papers? I would recommend that the authors signify the novelty of the present research work more effectively.
Experimental procedures
Figure 1: the authors may add scale bars for the images shown in Fig. 1. By adding scale bars, readers can understand the degree of magnification and the images will be comparable with other studies.
Line 64: There is an abbreviation “FSM” what does this term mean? Did the authors mean “FSW”?
Line 69: The terms “EPMA” and “XRD” should be introduced in the text.
Line 73: I would call it crosshead speed not loading speed.
In “Experimental Procedure”, it is better that the authors mention the working temperature at the beginning of the welding process (for example if the welding process was carried out in room temperature or 20 oC).
Results and Discussion
It is important and necessary to show the level of grain refinement due to the severe plastic deformation in FSW processing. The authors may add the microstructure and the grain sizes of some selected joints (optical microscopy, BSE or EBSD techniques can be used to display microstructure).
Figure 2: There is no scale bars in this figure.
Figure 2: There is no scale bars in this figure.
Figure 8: I suggest that the elongation to failure will be demonstrated since tensile testing provide not only the strength of samples but also elongation and ductility.
Line 177-180: The analysis of fracture surfaces should be connected to the obtained ductility of samples.
Round 2
Reviewer 1 Report
In my opinion, the authors have not made enough to improve the manuscript's clarity and readability as well as novelty. Many of my comments were not addressed to at all despite their importance. On the contrary, those were addressed, received unsatisfying expanations. Also despite providing the replies to comments no corrections were made to the text.
The objective of this paper was not formulated. Still lacking information about equipment manufacturers.
Reviewer 4 Report
Due to the modifications made in the manuscript, the current version of paper is suitable for publication.
Round 3
Reviewer 1 Report
More sophisticated explanations can be provifed with respect to plasticized metal deformation. It is still sounds arguable for me that "... Mg alloy has relative lower ductility compared with Al alloy due to its HCP structure, ...".
Mg and Al have their meling points at 650 and 660 C , respectively, while mean temperature in FSW zone is usually about 400-450C. Being deformed and stirred at these temperatures both metals are nanocrystalline grains and therefore their crystalline lattices are of much less importance for their deformation as compared to the grain boundary slip mechanism.
Also heat is generated from intermetallic compound reaction between Al and Zn which makes the metal flow even easier.
Metal quasi-viscous flow in FSW can not explain the layer-by-layer or "onion ring" structures often formed in the weld core. Therefore, adhesion transfer mechanism must be at least mentioned while discussing the weld formation, in particularly the filling of the zone behind the moving FSW tool.
